# Learning to Keep a Promise: Scaling Language Model Decoding Parallelism with Learned Asynchronous Decoding

Tian Jin [* 1 †]   Ellie Y. Cheng [* 1]   Zachary Ankner [1]   Nikunj Saunshi [2]   Blake M. Elias [3]   Amir Yazdanbakhsh [4]
Jonathan Ragan-Kelley [1]   Suvinay Subramanian [3]   Michael Carbin [1]

## Abstract

Decoding with autoregressive large language models (LLMs) traditionally occurs sequentially, generating one token after another. An emerging line of work explored parallel decoding by identifying and simultaneously generating semantically independent chunks of LLM responses. However, they rely on hand-crafted heuristics tied to syntactic structures like lists and paragraphs, making them rigid and imprecise. We present PASTA, a learning-based system that teaches LLMs to identify semantic independence and express parallel decoding opportunities in their own responses. At its core are PASTA-LANG and its interpreter: PASTA-LANG is an annotation language that enables LLMs to express semantic independence in their own responses; the language interpreter acts on these annotations to orchestrate parallel decoding on-the-fly at inference time. Through a two-stage finetuning process, we train LLMs to generate PASTA-LANG annotations that optimize both response quality and decoding speed. Evaluation on AlpacaEval, an instruction following benchmark, shows that our approach Pareto-dominates existing methods in terms of decoding speed and response quality; our results demonstrate geometric mean speedups ranging from 1.21× to 1.93× with corresponding quality changes of +2.2% to -7.1%, measured by length-controlled win rates against sequential decoding baseline.

*Equal contribution ,†Part of this work was completed during an internship at Google.  [1]MIT CSAIL, Cambridge, USA [2]Google Research, New York, USA [3]Google, Mountain View, USA [4]Google DeepMind, Mountain View, USA. Correspondence to: Tian Jin <tianjin@csail.mit.edu>, Michael Carbin <mcarbin@csail.mit.edu>.

*Proceedings of the $42^{nd}$ International Conference on Machine Learning*, Vancouver, Canada. PMLR 267, 2025. Copyright 2025 by the author(s).

## 1. Introduction

Autoregressive decoding is a fundamental efficiency bottleneck in large language model (LLM) inference. Contemporary LLMs routinely require multiple seconds or even minutes of decoding time to complete user requests (Jiang et al., 2024; Touvron et al., 2023; Jaech et al., 2024; Guo et al., 2025). This latency stems from the sequential nature of autoregressive decoding, which leads to inefficient hardware utilization during inference. While training achieves 40-60% Model Flops Utilization (MFU) (Korthikanti et al., 2022), inference typically achieves less than 20% MFU (Pope et al., 2022).

**Semantic Independence.** Recent works like Skeleton-of-Thought (SoT) and APAR leverage semantic independence in LLM responses as a source of parallelism, decoding independent chunks of tokens in parallel. Namely, these methods decode semantically independent *chunks* (contiguous sequences of tokens) of tokens in the response in parallel. Given a request, SoT first produces a bullet-point outline, then applies regular-expression-based syntactic pattern matching to extract points that are then expanded in parallel. APAR, in contrast, applies regular-expression-based syntactic pattern matching on training data to identify structures like lists and paragraphs, and finetunes an LLM to decode in parallel the item descriptions given list items and the paragraph bodies given the first sentences.

While semantic independence broadly exists in LLM responses, relying purely on syntactic heuristics – manifested as hand-crafted regular expressions – to identify them has limitations. First, these heuristics lack scalability, requiring manual engineering to capture more semantic independence, even as more training compute becomes available. Second, they lack robustness, failing to detect semantic independence when responses deviate from expected patterns, even by a missing punctuation mark. These limitations motivate a learning-based approach to optimize LLMs' ability to identify semantic independence, enabling LLMs to find parallelization opportunities beyond fixed patterns.

**Learned Parallel Decoding.** We present PASTA, a system that teaches LLMs to identify and annotate parallelization

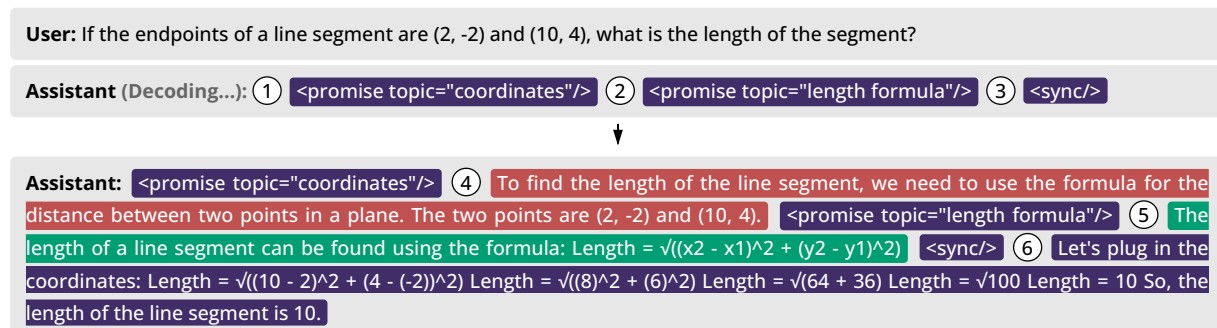

*Figure 1.* Example response from a PASTA model executed by the PASTA-LANG interpreter. The interpreter begins with only the main thread. It first decodes ①, and it creates an asynchronous decoding thread, which decodes ④ in red. In parallel, the main thread decodes ②. It creates another asynchronous decoding thread, which contains both the `<promise/>` tag on coordinates extraction and the `<promise/>` tag on length formula in its prefix, and decodes ⑤ in green. The main thread continues decoding in parallel to both threads to get ③. It wait at this point until all other threads complete. The interpreter then inserts each asynchronous content after their corresponding `<promise/>` tags. Finally, the interpreter decodes ⑥, with both of the asynchronously decoded content in the prefix.

opportunities in their own responses. Our system consists of a set of annotations that extend the model's vocabulary for asynchronous decoding, an interpreter that acts on these annotations to orchestrate parallel decoding, and a finetuning procedure that optimizes LLMs' ability to identify and express parallelization opportunities. Through this system, LLMs develop and execute their own asynchronous decoding strategies. In Figure 1, we show how these components implement asynchronous decoding.

**Annotations.** Our annotation language, PASTA-LANG (PArallel STructure Annotation LANGuage), enables LLMs to express semantic independence in their responses. In Figure 1, we show a PASTA-LANG-annotated response. The `<promise/>` tags serve as placeholders for content chunks that are semantically independent to each other, such as extracting coordinates (Tag ①) and recalling the line segment length formula (Tag ②). Each `<promise/>` tag includes a `topic` attribute that concisely describes the chunk. When further decoding steps require conditioning on tokens that are still being asynchronously decoded, the LLM issues an `<sync/>` tag to indicate so, as shown at ③ in Figure 1.

**Interpreter.** We develop the PASTA-LANG interpreter, which acts on PASTA-LANG annotations to orchestrate asynchronous decoding during inference. It launches parallel decoding threads for semantically independent contents marked with `<promise/>` tags and synchronizes them at `<sync/>` tags. The interpreter simultaneously decodes multiple non-contiguous token chunks from the LLM, improving overall decoding latency.

**Finetuning.** Training an LLM to generate PASTA-LANG annotations starts with two manual inputs: seven human-crafted demonstrations and a description of the PASTA-LANG annotation language. Prompting the Gemini 1.5 Flash model (Anil et al., 2024) with these manual inputs, the PASTA system initiates an automated two-stage finetuning process. In the first stage, PASTA uses the prompted Gemini

model to create the *Pasta-SFT* dataset by annotating the SlimOrca instruction-finetuning dataset (Lian et al., 2023b) with PASTA-LANG annotations that identify semantically independent chunks compatible with asynchronous decoding. PASTA then finetunes an LLM on this dataset to produce a model that generates PASTA-LANG annotations.

In the second stage, PASTA creates another dataset by sampling the finetuned LLM and scoring each output based on its quality and latency. Unlike traditional uses of preference optimization to improve response quality (Gui et al., 2024; Rafailov et al., 2023), we adapt one such algorithm for PASTA to optimize for both output quality and latency. PASTA applies preference optimization to the finetuned LLM on this dataset to produce a model with improved output quality and latency. This second stage of finetuning features a quality weight hyperparameter that controls the trade-off between quality and speedup. Through repeated iterations of the second stage, PASTA creates models that respond with increasingly better quality and lower latency.

**Results.** Varying the quality weight hyperparameter, PASTA produces a suite of models with different quality-latency trade-offs. We evaluate these models on 805 representative instruction-following prompts from AlpacaEval (Li et al., 2023; Dubois et al., 2024). After one iteration of preference optimization, these models Pareto-dominate all existing asynchronous decoding methods. Additional iterations of preference optimization further improve the speedup-quality Pareto frontier, showing no signs of saturation even after two iterations. Our results demonstrate geometric mean speedups ranging from 1.21x to 1.93x[1] with corresponding quality changes of +2.2% to -7.1% respectively, measured as in length-controlled win rates

---

[1]Geometric mean should be used to compute normalized values (Fleming & Wallace, 1986). However, the prevailing practice in parallel decoding literature uses arithmetic averaging when reporting speedup, which would show this result as 1.57-2.6×.

**Contribution.** We present a collection of contributions:

- We design PASTA-LANG to be an annotation language that enables LLMs to annotate semantically independent chunks of tokens in their own responses.

- We implement a PASTA-LANG interpreter that efficiently orchestrates asynchronous decoding based on PASTA-LANG annotations at inference time.

- We develop a two-stage finetuning technique that trains LLMs to identify diverse patterns of semantic independence in their output and express them through PASTA-LANG annotations, while directly optimizing for both response quality and inference speedup.

- We evaluate our method on AlpacaEval (Li et al., 2023; Dubois et al., 2024), a suite of 805 representative instruction-following prompts, and find our method Pareto-dominate all existing asynchronous decoding methods in terms of quality and speedup.

**Implication.** PASTA demonstrates the utility of incorporating latency objectives into the standard preference optimization step during LLM post-training. The effectiveness and scalability of this approach makes it a practical prescription for reducing LLM decoding latency.

## 2. Asynchronous Decoding

To provide context for how PASTA-LANG relates to other parallel decoding techniques for accelerating LLM decoding, we present a dichotomy of parallel decoding techniques. Specifically, a given parallel decoding strategy can be categorized as either performing *synchronous* or *asynchronous* decoding. In synchronous decoding, only a single chunk is decoded in parallel while the rest of the generation is halted. In contrast, during asynchronous decoding, multiple chunks of of the language model's output are decoded independently in parallel.

We consider speculative decoding (Leviathan et al., 2023; Chen et al., 2024; Stern et al., 2018; Cai et al., 2024; Ankner et al., 2024; He et al., 2024; Fu et al., 2024; Spector & Re, 2023; Santilli et al., 2023) as a prototypical example of synchronous decoding. It decodes multiple tokens within a single chunk in parallel, but must complete that chunk before moving on to any subsequent tokens.

In contrast, works such as SoT (Ning et al., 2024) and APAR (Liu et al., 2024) implement asynchronous decoding techniques. Both methods enable decoding to jump ahead in the output sequence and generate tokens before previous positions are filled, resulting in multiple chunks of the output being decoded in parallel. While our PASTA system also implements asynchronous decoding, we improve upon previous works by employing a learning-based system to identify parallelization opportunities, instead of relying on human-defined heuristics. By training an LLM to identify and exploit parallelization opportunities, PASTA achieves Pareto-optimal trade-off between speedup and quality as compared to previous asynchronous decoding techniques.

## 3. Language and Interpreter Design

PASTA-LANG is an XML-like annotation language designed for a language model to annotate semantic independence in its own response. We present the syntax of the language and the operations of the interpreter in this section.

**Syntax.** PASTA-LANG defines three tags: `<async>` tags which appear in pairs to wrap around blocks of content, and two standalone tags `<promise/>` and `<sync/>`. A `<promise/>` tag requires two attributes: a string attribute `topic` and an integer attribute `tokens`, and must appear before the content block it refers to.

**Interpreter.** A PASTA-LANG-equipped language model initiates asynchronous decoding by generating the PASTA-LANG tags that the PASTA-LANG interpreter executes to implement asynchronous decoding. We describe here the functionality of each tag and how they are used by the interpreter. Figure 2a shows how the interpreter orchestrates asynchronous decoding.

With the user query Ⓐ as the prefix, the interpreter decodes sequentially until encountering a `<promise/>` tag at tag Ⓑ. The `topic` attribute indicates the topic of the chunk that will be decoded asynchronously, and the `tokens` attribute estimates the number of tokens in multiples of 10 in the `<async>` tag. These attributes provide context for decoding future tokens. The interpreter then initiates a new asynchronous decoding thread named "Fork#1". The main thread continues decoding in parallel, while the new thread first appends an `<async>` tag to its prefix at Ⓒ and then decodes content matching the specified `topic` until reaching a `</async>` tag at Ⓓ. The main thread proceeds without conditioning on any asynchronously decoded content. When encountering a `<sync/>` tag at Ⓔ, the interpreter pauses to wait for all asynchronous decoding threads to complete, and enables the language model to condition on asynchronously decoded content for decoding subsequent tokens in Ⓕ.

**Efficiency.** Our interpreter implementation addresses several key challenges in efficient asynchronous decoding, with KV-cache management being the core issue. Since ML compilers often requires static tensor shapes for effective compilation (Sabne, 2020; Paszke et al., 2019), we assume a fixed batch size and sequence length [2] Implementing asynchronous decoding naively as batched decoding faces two suboptimal options: 1) allocate differently sized KV-cache

---

[2] We use a batch size of 1 and max sequence length of 2048 as in Liang et al. (2025).

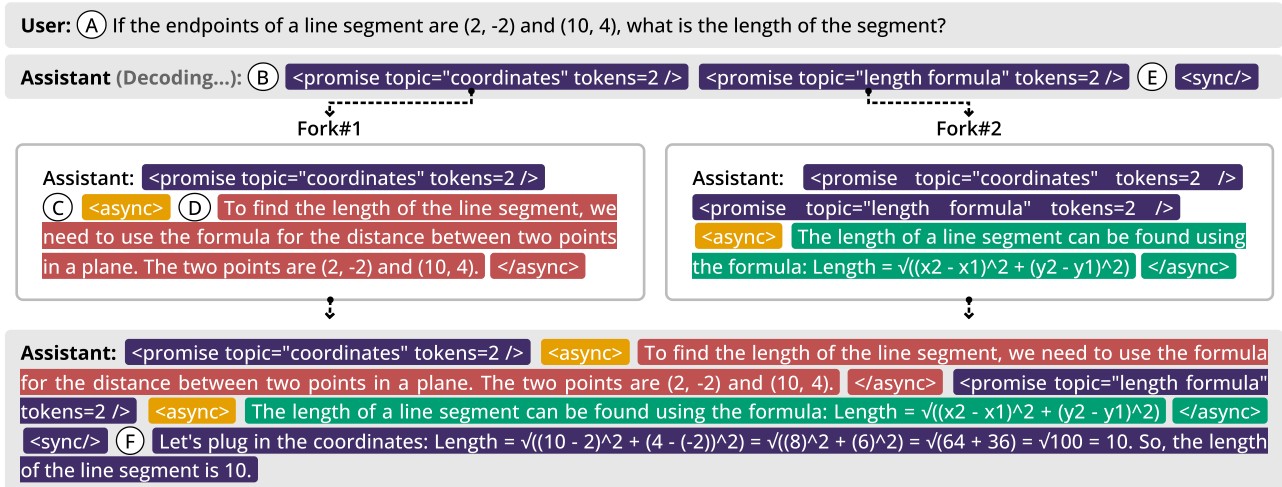

(a) PASTA-LANG interpreter orchestrates parallel decoding. (A) shows the user prompt. (B) shows the `<promise/>` tag which initiates the first asynchronous decoding thread named "Fork#1". (C) indicates where the interpreter appends an `<async>` tag to the prefix of Fork#1, signaling Fork#1 should complete the promised content with topic "coordinates". (D) denotes the asynchronous generation by Fork#1. (E) shows the `<sync/>` tag where the interpreter pauses to wait for all asynchronous generations. (F) shows the main thread decodes the remaining content with both asynchronous generations in its prefix.

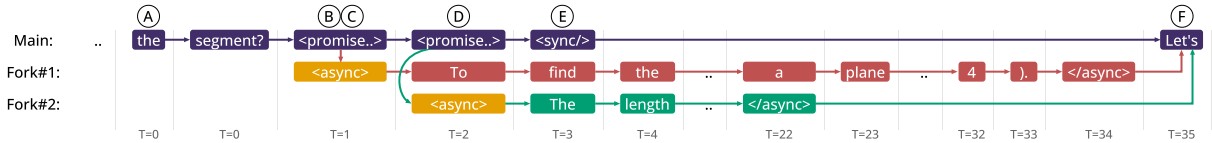

(b) Decoding parallelism and attention patterns at each timestamp. At each timestamp, we show the tokens decoded in parallel in that timestamp. The directed edges between tokens show the attention relationships: each edge connects a token to the very next token that may attend to it. A token can then attend to any tokens that can be reached by following these edges back through the graph. (A) shows the last few tokens of the user query. (B) shows when the interpreter decodes a `<promise/>` token, after which it immediately appends an `<async>` token for Fork#1 at (C). Subsequently at (D), Fork#1 begins asynchronous decoding, while in parallel, the interpreter creates another decoding thread (Fork#2). At (E), when encountering the `<sync/>`, the interpreter pauses the main thread until all asynchronous threads complete. Finally at (F), the main thread resumes decoding with both asynchronously decoded content in its prefix.

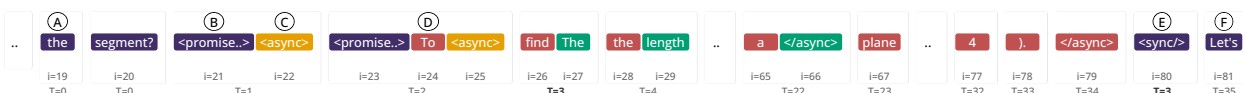

(c) KV-cache layout throughout parallel decoding. At (A), the last few tokens of the user prompt, the KV-cache is laid out in typical contiguous manner. Starting at (B), the KV-cache begins to interleave between threads, while inserting the corresponding `<async>` token for the new thread at (C). (D) shows parallel decoding in progress, with tokens from Fork#1 being generated while the main thread continues decoding. At (E), the `<sync/>` token (decoded by the main thread at T=3) is inserted into the KV-cache. After synchronization, the KV-cache returns to a contiguous layout at (F).

*Figure 2.* Details for efficient PASTA-LANG interpreter implementation. Color shows the identity of the decoding thread (purple=main, red=Fork#1, green=Fork#2); orange denotes interpreter-inserted tokens.

pools and switch between them upon thread creation and termination, which wastes precious accelerator memory, or 2) pre-allocating a fixed number of decoding threads, leading to wasted memory and computation due to inactive threads. We further discuss the drawbacks of naive option 2) in Appendix B. Instead, we store KV-cache from all asynchronous decoding threads in a single contiguous pre-allocated memory pool with an interleaved layout.

Figures 2b and 2c illustrates our approach. We denote decoding timestamp with T and index in the KV-cache pool with i. When only one thread is active at T=0 (Ⓐ), it appends new KV-cache sequentially to the pool at i=19-20. When the main thread (purple) decodes a `<promise/>` token at T=1 (Ⓑ), the interpreter immediately appends an `<async>` token to signal the start of a new thread Fork#1 (Ⓒ). At T=2 (Ⓓ), Fork#1 begins asynchronous decoding, while the interpreter initiates another asynchronous decoding thread Fork#2. At T=3, (Ⓔ), the interpreter decodes 3 tokens in parallel from 3 active threads, where the `<sync/>` token signals pausing to wait for other threads to complete and therefore this token does not enter into the KV-cache pool until later. Where as the other two tokens do enter the KV-cache pool in neighboring positions (i=26-27). The two threads (green/red) continue to decode in parallel, alternating as they append tokens to the KV-cache pool (T=3-22).

To prevent cross-thread interference with this interleaved KV-cache layout, we use attention masks to ensure threads cannot attend to each other's tokens before synchronization. In Figure 2b, each directed edge connects a token to the very next token that may attend to it. A token can attend to any ancestor token reachable by following these edges backward through the graph. A token in an asynchronous thread can only attend to tokens within its own thread and tokens from the main thread that existed before the thread was spawned. For example, the token `find` (at T=3, red) attends to `To` (at T=2, red) and `segment?` (at T=0, purple), but cannot attend to the second `<async>` token (at T=2, orange), the second `<promise/>` token (at T=2, purple), or the `<sync/>` token (at T=3, purple).

Once the interpreter decodes the `<sync/>` token in the main thread at T=3 (Ⓔ), it pauses the main thread to synchronize, waiting for both forks to complete: Fork#1 and Fork#2 decode their `</async>` tokens at T=34 and T=22 respectively. After the wait is over at T=34, the interpreter inserts the `<sync/>` token into the KV-cache pool at i=80, and the main thread resumes decoding (Ⓕ) while conditioning on both forks' asynchronous generations.

While prior works like radix attention (Zheng et al., 2023) enables multiple decoding threads to share attention to a common prefix, our design has to additionally address the challenge of enabling a single decoding thread to attend to multiple asynchronously decoded threads.

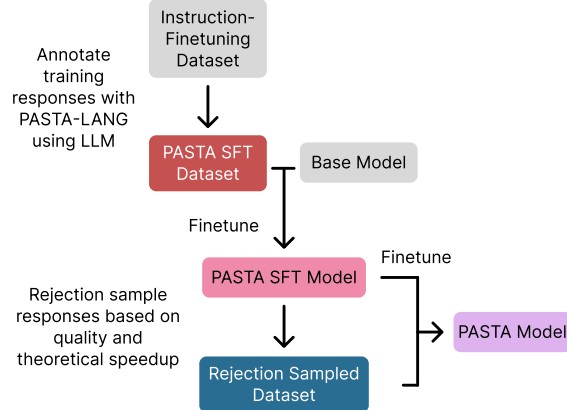

*Figure 3.* PASTA-LANG dataset creation and model training.

# 4. Training PASTA-LANG Capable Models

We present a two-stage finetuning process that trains an LLM to annotate semantic independence in their own responses using PASTA-LANG. Figure 3 illustrates the PASTA system pipeline for dataset construction and model training to produce a PASTA-LANG-equipped model.

**Building a *Pasta-SFT* Dataset.** The PASTA system pipeline begins by constructing an initial finetuning dataset, which we refer to as the *Pasta-SFT* dataset. It prompts the Gemini 1.5 Flash (Anil et al., 2024) to add PASTA-LANG annotations to responses from an instruction-following dataset We provide Gemini 1.5 Flash with 7 human annotated examples and a description of the syntax and semantics of PASTA-LANG and have it label a 100K response subset of the SlimOrca dataset (Lian et al., 2023a)[3]. We provide the annotation prompt used in Appendix A.

**Training a *Pasta-SFT* Model.** PASTA then finetunes the base LLM on the *Pasta-SFT* dataset, producing what we call the *Pasta-SFT* model. Since the model must decode content after `<promise/>` tags without access to the corresponding `<async>` content, we implement three key modifications to the standard next-token prediction finetuning algorithm.

First, the attention mask prevents the tokens after a `<promise/>` from attending to content within its `<async>` tags until a `<sync/>` tag is reached. This enforces the semantic independence constraint between asynchronous blocks. Second, during inference, the position IDs of tokens after a `<promise/>` cannot be known until fully decoding the corresponding `<async>` block. To handle this, we train the model to predict the length of each `<async>` block in the `tokens` attribute of its `<promise/>` tag. This prediction enables us to assign estimated position IDs to tokens that follow the `<promise/>` tag, even before fully decoding the `<async>` block. Finally, to enable the model to continue

---

[3]Due to API limitations, we only successfully annotated 87K/100K instruction-response pairs as the *Pasta-SFT* dataset.

decoding past `<promise/>` tags, we set the next-token prediction target at each `<promise/>` to be the first token after its corresponding `<async>` block. This enables the model to skip past placeholder `<promise/>` tags.

**PASTA-LANG Preference Optimization.** PASTA further improves the *Pasta-SFT* model with a second stage of training that directly optimizes for quality and speedup achieved by the PASTA-LANG annotations.

First, PASTA trains a baseline sequential model *Baseline-SFT* by fine-tuning the same base model on *Pasta-SFT* dataset without the annotations. Then, for each prompt in the *Pasta-SFT* dataset, PASTA samples $N$ different PASTA-LANG annotated responses from the *Pasta-SFT* model using a temperature $T$. It then scores each of the $N$ sampled responses. We set $N = 10$ and temperature $T = 1$. The score for ranking the sampled responses is a combination of:

1. Response *theoretical speedup* – the ratio between (1) the total number of tokens in *Baseline-SFT*'s response, and (2) the length of the longest sequence that must be decoded sequentially for *Pasta-SFT*'s response. This measures the maximum achievable speedup of *Pasta-SFT* over *Baseline-SFT*.

2. Response *quality* – for each sampled response, PASTA computes a confidence-weighted win-loss ratio by comparing it against the *Baseline-SFT*'s response and the original SlimOrca response. Each comparison appears in both orders and is judged by Gemini 1.5 Pro, which provides a preference and a probability for that preference between 0 and 1, interpreted as confidence. The final ratio is the confidence-weighted sum of wins divided by the confidence-weighted sum of losses.

Each sampled response receives a score: speedup $+ \lambda \times$ quality, where $\lambda$ is the *quality weight*. For each prompt, PASTA selects the highest and lowest-scoring response as the preferred and un-preferred example, respectively. We explore alternative scoring methods in Appendix D.

While our methodology is compatible with any LLM preference optimization algorithm, in this work we use BoNBoN optimization (Gui et al., 2024) for it was the state-of-the-art algorithm at the time of writing this paper. The BonBon algorithm trains a model to approximate the best-of-N response distribution by combining an SFT loss on the preferred example with an IPO preference loss (Azar et al., 2024) between the best and worst response. Specifically, the BoNBoN objective is:

$$\mathcal{L}_{\text{BoNBoN}}(\theta; D, \theta_{\text{init}}) = \mathbb{E}_{x,y^+,y^- \sim D}[-\alpha \log p_\theta(y^+|x)$$
$$+ (1-\alpha)(\log \frac{p_\theta(y^+|x)}{p_\theta(y^-|x)}$$
$$- \log \frac{p_{\theta_{\text{init}}}(y^+|x)}{p_{\theta_{\text{init}}}(y^-|x)} - \frac{1}{\beta})^2]$$

where $\theta$ are the model weights being trained, $\theta_{\text{init}}$ are the initial model weights, $x$ is a prompt, $y^+, y^-$ are the best and worst-of-N PASTA-LANG-annotated responses respectively, and $\alpha$ is a hyperparameter to weight the SFT and IPO loss contributions. We set $\theta_{\text{init}}$ to be the *Pasta-SFT* model. We use a learning rate of 5E-7 and set $\alpha$ to 0.005 as recommended by (Gui et al., 2024).

# 5. Experiment

We evaluated the performance of our PASTA-LANG-equipped model in terms of speedup, parallelism, and response quality and show that it is Pareto-optimal compared to other asynchronous decoding techniques.

## 5.1. Experimental Setup

**Baselines.** We present the following baselines: standard autoregressive decoding from *Baseline-SFT*, APAR decoding (Liu et al., 2024) and SoT decoding (Ning et al., 2024). APAR and SoT are examples of asynchronous decoding techniques relying on hand-crafted syntactic heuristics. We evaluate sequential autoregressive decoding using the *Baseline-SFT* model finetuned on the the *Pasta-SFT* dataset without PASTA-LANG annotations. To evaluate APAR decoding, we train an APAR model again on the same *Pasta-SFT* dataset, except preprocessed following the official APAR methodology. Using regex and filters, we recreated the APAR heuristics for extracting structured data as described in their work (Liu et al., 2024).

**Models and Hyperparameters.** For all experiments, we use Gemma 7B (Mesnard et al., 2024) as the base model. We finetune all models using a batch size of 8, a learning rate that decays linearly from 1e-5 to 0, and train for a total of 4 epochs. We chose these hyperparameters as they maximized the quality of the baseline model. [4] We provide further details on hyperparamter selection in Appendix C.

There are two PASTA specific hyperparamters: $\lambda$, the quality weight used for building preference pairs (Section 4); and $r$, the number of BoNBoN preference optimization iterations. We train multiple PASTA models, with $\lambda = 1, 2, 4, 8$ and set $r = 2$. We refer to each model as *Pasta-BoN-$\lambda$*. We also train PASTA models while optimizing exclusively for quality, denoted as *Pasta-BoN-$\infty$*.

**Hardware and Software.** We evaluate decoding performance using PyTorch with torch.compile optimization set to maximum auto-tuning mode (Paszke et al., 2019). All experiments run on H100 GPUs using greedy decoding. We

---

[4]For reference, our baseline model achieves a 38% length controlled win rate against Gemma-7B-it, Google's officially released instruction-tuned Gemma-7b model. This comparison demonstrate that despite our smaller-scale experimental setup, our baseline model performs competitively against state-of-the-art model.

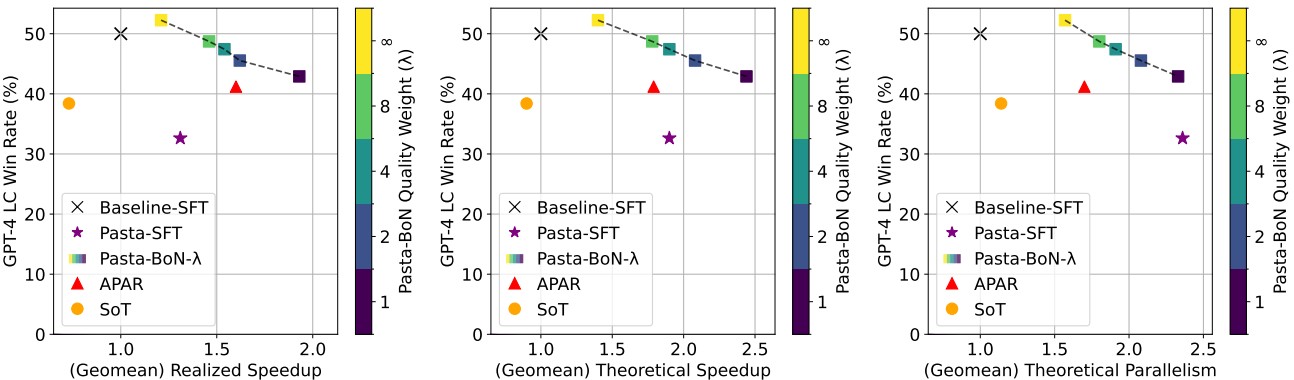

*Figure 4.* Left (Realized Speedup). PASTA models achieved Pareto-optimal quality-speedup trade-off than asynchronous decoding strategies with hand-crafted heuristics. Middle (Theoretical Speedup). The realized speedup using PASTA-LANG interpreter is close to the theoretical speedup. Right (Theoretical Parallelism).PASTA responses show high degree of parallelism.

use a batch size of 1 as is common with parallel decoding literature (Leviathan et al., 2023). To avoid measuring compilation overhead, we decode each request twice and only take the timing of the second of two decoding runs.

**Evaluation.** We evaluate all models on the AlpacaEval benchmark (Dubois et al., 2024; Li et al., 2023), an open ended suite of 805 representative prompts. For each decoding method we evaluate, we compute the following metrics:

1. We measure the *realized speedup* against the baseline as the ratio between the wall-clock decoding times: baseline model time divided by test model time, both using the PASTA-LANG interpreter.

2. We calculate the *theoretical speedup* against the baseline as the ratio between (1) the total number of tokens in the baseline response and (2) the length of the longest sequence that must be decoded sequentially in the test model's response, as described in Section 4.

3. We measure the *theoretical parallelism* in the model output: the ratio of (1) the total number of non-control tokens to (2) the length of the longest sequence of tokens that must be decoded sequentially.

4. We measure the *quality* as the length-controlled, LLM-as-a-judge win-rate using AlpacaEval benchmark (Li et al., 2023; Dubois et al., 2024) when compared to the baseline model. We use Gemini 1.5 Pro (Anil et al., 2024) as the judge model for development and GPT4 as the judge for evaluation, to prevent reward hacking.

We aggregate the speedup and parallelism over each prompt in the dataset using the geometric mean[5].

---

[5]The arithmetic mean of ratios can lead to inconsistent results depending on the baseline used to compute the ratio and is therefore not appropriate for our use (Fleming & Wallace, 1986). However, the prevailing practice in parallel decoding literature uses

## 5.2. Results

The left, middle and right plots in Figure 4 show how response quality trades off against realized speedup, theoretical speedup, and parallelism respectively across PASTA and baseline models. We mark the baseline (50% win rate with no speedups) with an X. An ideal asynchronous decoding strategy should match this baseline in quality while surpassing it in speedup; the closer to the top right corner of the plot, the better the technique.

**Pareto-Optimality.** The left plot in Figure 4 shows that PASTA models achieve Pareto-optimal trade-off between quality and realized speedup. The best-quality model (*Pasta-BoN-∞*) achieved 52.3% win rate at 1.21x speedup, whereas the best-speedup model (*Pasta-BoN-1*) achieved 42.9% win rate at 1.93x speedup. The *Pasta-BoN-2* model Pareto-dominate all prior asynchronous decoding techniques, achieving superior quality-speedup trade-off.

The APAR model and *Pasta-BoN-2* achieved similar speedups (1.6x vs 1.62x), but *Pasta-BoN-2* obtained a 4.3% higher win rate. This result stems from APAR's reliance on syntactic heuristics for identifying semantic independence, which can lead to false positives. Similarly, while APAR and *Pasta-BoN-1* showed comparable win rates (41.2% vs 42.9%), *Pasta-BoN-1* delivered a 20.6% higher speedup. This superior speedup stems from the flexibility of PASTA-LANG's annotation, which enables asynchronous decoding at any position in the output.

Notably, we do not observe any speedup by SoT (Ning et al., 2024), when applied to *Baseline-SFT*. We believe that SoT, as a prompt-based method, requires the base model to have strong instruction-following ability to perform well, and

---

arithmetic mean when reporting speedup. We provide the speedup computed with arithmetic mean for reference in Appendix E.

validated this hypothesis by applying SoT to the stronger official instruction-finetuned Gemma-IT model from Google. With this generous implementation, SoT achieves a 1.61x speedup while dropping its win rate by 12%. In contrast, our *Pasta-BoN-2* achieves 1.62x speedup with only a 5% drop to win rate.

**Role of** $\lambda$. We observe that the quality weight $\lambda$ serves as an effective control knob for the trade-off between quality and speedup. A lower weight results in more aggressive optimization for speedup at the cost of reduced quality.

**Theoretical speedup.** Comparing the left (realized speedup) and middle (theoretical speedup) plot in Figure 4 shows that the combination of PASTA-LANG interpreter and PASTA models deliver realized speedup close to theoretical optimal.

**Parallelism.** The right plot in Figure 4 shows that PASTA models achieve a high degree of theoretical parallelism. Notably, while the *Pasta-SFT* model starts with high theoretical parallelism, this does not translate into significant decoding speedup. This disconnect occurs because the induced parallelism is not effective - the model learns to generate redundant content, which hurts both quality and speedup. Preference optimization effectively mitigates this pathology, as the *Pasta-BoN* models deliver speedups commensurate with their theoretical parallelism.

**Conclusion.** PASTA is an effective technique for enabling asynchronous decoding, producing Pareto-dominant performance. PASTA also enables flexible trade-off between speedup improvements and response quality.

## 6. Sensitivity Analysis.

In this section, we investigate how three key design choices of the PASTA system impact model quality and latency: a) the number of preference optimization iterations, b) the configurations of positional embeddings, and c) the scoring method for PASTA-LANG annotated responses.

### 6.1. Number of Iterations

We examine how the number of preference optimization iterations affects the quality-speedup trade-off in PASTA models; we observe continuous improvements as training compute increases and find distinct optimization dynamics at different preference optimization iterations.

**Methodology.** We analyzed the impact of preference optimization on the speedup-quality trade-off by PASTA models at 5 different stages: initial (aka *Pasta-SFT*), 10% into Round 1, and 100% into Round 1, 10% into Round 2, and 60% into Round 2[6]. The initial stage is represented by a single star, while results from later stages contain four points each, corresponding to quality weights ($\lambda$) of 1, 2, 4, and

[6]We stopped at 60% of Round 2 due to time constraints.

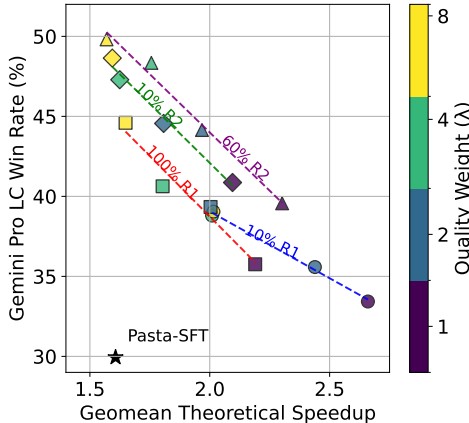

*Figure 5.* Increasing the number of preference-optimization rounds steadily improves the quality–latency trade-off.

8. We visualize each stage's data using distinct colors in Figure 5. To illustrate the difference in quality-speedup trade-off between stages, we computed linear fits for each stage after *Pasta-SFT* and plotted the linear fit using the same color as each stage.

**Results.** Figure 5 demonstrates the scalability of preference optimization, as increased training compute continuously improves the Pareto frontier toward better quality-speedup trade-offs (i.e., top right corner). As is common when scaling LLMs with more training compute (Kaplan et al., 2020), we observe diminishing returns, though we do not observe saturation after two rounds of preference optimization.

Observing the linear fits of quality-speedup trade-off within each stage reveals distinct optimization dynamics. The initial 10% of Round 1 preference optimization aggressively optimizes for speedup, moving the group to the right of the plot. However, from 10% Round 2 to 60% Round 2, we observe an emphasis on quality improvements, shifting the group upward. Overall, preference optimization effectively navigates the trade-off space by exploring both quality improvement and speedup improvement.

**Conclusion.** Preference optimization is a scalable technique that improves the speedup-quality trade-off with increased training compute. Unlike prior asynchronous decoding techniques that require hand-crafted syntactic heuristics, our technique directly converts computational resources into better quality and speedup trade-off, offering a more scalable path to improving decoding speed.

### 6.2. Positional Embedding

Asynchronous decoding methods introduce uncertainty over the true position of tokens as they are being generated as the output is generated in a non-sequential manner. Namely, the main decoding thread is unaware of the true length of of any previously occurring `<async>` blocks that have not been synchronized. Thus, the main thread must estimate

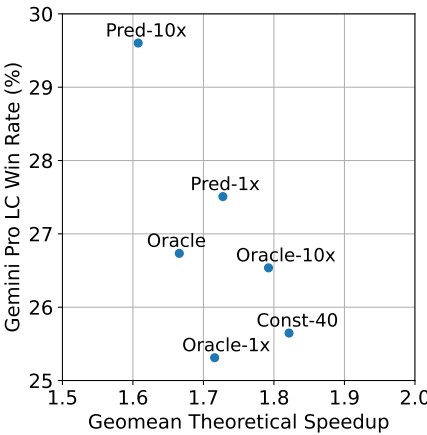

*Figure 6.* Predicting position IDs in multiples of ten (Pred-10x) gives the best quality with minimal speed-up loss.

the number of tokens in an `<async>` block before continuing generation. If the predicted number of tokens for the `<async>` block does not match the true number of tokens generated, this can lead to errors as the position IDs after synchronization will either not increase monotonically (predicted too few) or contain a gap (predicted too many).

**Methodology.** To minimize the error between the true and predicted position IDs, we compare three different approaches for position ID assignment:

*Fixed-length*: We make the assumption that each `<async>` block has a fixed length. In our experiments, we chose this length to be forty tokens as this is slightly larger than the median `<async>` block length in our training data.

*Length Prediction*: We train the model to predict the length of each `<async>` block, and then use the model's predictions during decoding. We evaluate two variants that predict `<async>` block lengths at different granularities: 1) Pred-1X: Predict the `<async>` token length exactly; and 2) Pred-10X: Predict the `<async>` token length as a multiple of ten for coarser granularity.

*Oracle*: We use the ground truth length of each `<async>` block (i.e. the position IDs of the tokens in the block if the chunk was decoded sequentially) to assign position IDs. While the oracle position IDs are infeasible to obtain during deployment, we evaluate the performance of oracle ID decoding to serve as a reference point for the performance of decoding with no error in the position ID calculations. We evaluate two different granularity for the oracle position IDs: 1) Oracle-1X: Using the true `<async>` block length; and 2) Oracle-10X: Using the true `<async>` block length rounded to the nearest multiple of ten. We consider one final oracle baseline, which we refer to as Oracle, where the exact `<async>` block lengths are used to offset the position IDs but the length is not included in the `<promise/>` tag.

**Results.** Figure 6 presents both the response quality and

speedup for models finetuned on the *Pasta-SFT* dataset using each of the different position ID estimation techniques. Length prediction performs the best, achieving quality and speedup metrics matching (or even slightly exceeding) the Oracle. The Pred-10X variant offers a flexible solution to the position ID assignment problem that is admissible to preference optimization. Based on these results, we adopted Pred-10X as our position ID assignment strategy.

## 7. Related Work

After discussing parallel decoding approaches in Section 2, we now turn to other relevant research areas.

**Agent Planning/Tool Use.** Our work is related to the idea of agent planning and tool use (Yao et al., 2023; Schick et al., 2023; Shen et al., 2023; Liang et al., 2024; Lu et al., 2023). Prior studies show that LLM-based agents can solve complex tasks by planning and using tools such as web search and external APIs. Our work extends the suite of tools available to LLMs with the PASTA-LANG language and interpreter for improving their own decoding efficiency.

**Approximate Parallelization.** Our work extends the idea of approximate parallelization (Udupa et al., 2011; Misailovic et al., 2012). Udupa et al. (2011) proposed a framework that enables programmers to annotate breakable data dependencies in a program and developed a compiler and runtime that exploits these annotations to automatically parallelize otherwise sequential regions of code. Misailovic et al. (2012) opportunistically relaxes synchronization primitives in a parallel program to improve parallelism, to program outputs that are acceptably close to the original one. Similarly, we break the sequential decoding process of LLMs into approximately parallelizable and independent components and exploit the parallelism to improve decoding efficiency. Our work differs in that instead of relying on end-user annotation or compiler analysis, we teach LLMs to autonomously express parallelism in their own decoding process using the PASTA-LANG annotation language.

## 8. Conclusion

We present PASTA, a system that teaches LLMs to identify semantically independent chunks in their own responses and annotate them for parallel decoding opportunities. Through a set of annotations, an interpreter and a finetuning procedure, our method enables learned asynchronous decoding.

PASTA is an effective and scalable system which enables learned asynchronous decoding. Evaluation on AlpacaEval demonstrates that our approach Pareto-dominates existing asynchronous decoding methods in terms of quality and speedup. The improvements continue with additional training compute, showing no signs of saturation during our experimental evaluation.

## Impact Statement

This work improve the efficiency of LLM inference, which could lead to reduced computational resource requirements and improved accessibility to LLMs.

## Acknowledgement

This work was supported in-part by the Sloan Foundation and SRC JUMP 2.0 (CoCoSys). We are deeply grateful to William Brandon, Michael Brenner, Gintare Karolina Dziugaite, Xin Dong, Han Guo, Zhun Liu, Jesse Michel, Aniruddha 'Ani' Nrusimha, Ankit Singh Rawat, Narges Shahidi, Ananda Theertha Suresh, Xin Wang, Logan Weber, Cambridge Yang, and Xiaofan Zhang for their thoughtful feedback and detailed review of this manuscript. This work benefited significantly from the support of the Google CoreML Performance Team. We also thank the extended team at Google Research and Google DeepMind for their valuable contributions to this research.

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

# A. Dataset Annotation Prompt

Below is the prompt used to annotate our PASTA-LANG seed dataset. Additional in-context examples are omitted for presentation.

As a highly-paid expert annotator, you will be given a chatbot response, your job is to decide whether and how this response may be generated asynchronously in parallel by a large language model. Parallel text generation allows the model to generate segments of text simultaneously rather than sequentially, significantly improving efficiency and response times.

You will first identify whether the given chatbot response may be generated in parallel. You are to then annotate the chatbot response using specific tags that highlight segments suitable for parallel generation.

Use <async> tags to denote segments of text that may be generated asynchronously in parallel with respect to the text that follows. Thus apply <async> tags only to sentences that do not serve as necessary context for subsequent sentences. Sentences that are crucial for understanding or generating following text are not suitable for parallel asynchronous generation. For each <async> tag, include a very concise topic description of the text surrounded within the <async> tags. The topic description will be accessible to text generation after the closing async tag to ensure continuity and coherence.

Use the singleton <sync/> tag for synchronization. All content generated before <sync/>, including text marked by <async> is accessible to subsequent text generation after the <sync/> tag, ensuring continuity and coherence.

Detailed Instructions:
- Tagging Rules:
 - Use <async> tag in pairs.
 - Ensure that each <async> tag encompasses at least five words.
 - Refrain from altering the content of the response during annotation.
 - Use a maximum of 3 words in the topic description.
 - Use <sync/> sparingly as it introduces significant slowdown.

Input conversation:
```json
[
  {
    "from": "human",
    "value": "Article: In March 1968, CBS and Sony formed CBS/Sony Records, a

Japanese business joint venture. With Sony being one of the developers behind the compact disc digital music media, a compact disc production plant was constructed in Japan under the joint venture, allowing CBS to begin supplying some of the first compact disc releases for the American market in 1983.\n\nNow answer this question: In what year did CDs come to the American market?"
  },
  {
    "from": "chatbot",
    "value": "CDs (Compact Discs) came to the American market in 1983. This was made possible through a joint venture between CBS and Sony, named CBS/Sony Records, which was formed in March 1968. As Sony was one of the developers behind the compact disc digital music media, they built a compact disc production plant in Japan under this joint collaboration. This plant enabled CBS to supply some of the first compact disc releases for the American market in 1983. The introduction of CDs revolutionized the music industry as they provided a new and improved way to store and listen to music, providing higher sound quality and durability compared to the earlier vinyl records and cassette tapes."
  }
]
```
Annotated chatbot response:
```json
{
  "from": "chatbot",
  "value": "CDs (Compact Discs) came to the American market in 1983. <async topic='joint venture'>This was made possible through a joint venture between CBS and Sony, named CBS/Sony Records, which was formed in March 1968.</async> <async topic='japan production plant'>As Sony was one of the developers behind the compact disc digital music media, they built a compact disc production plant in Japan under this joint collaboration. This plant enabled CBS to supply some of the first compact disc releases for the American market in 1983.</async> <async topic='impact'> The introduction of CDs revolutionized the music industry as they provided a new and improved way to store and listen to music, providing higher sound quality and durability compared to the earlier vinyl records and cassette tapes.</async>"
}
```

...

## B. Naive Interpreter Implementation.

In this section, we illustrate the KV-cache layout for naive implementation before and after synchronization in Figure 7b and Figure 7c. We repeat the full model output in Figure 7a. Here we illustrate the inefficiencies associated with implementing asynchronous decoding as batched decoding, where we pre-allocate the KV-cache pool assuming a fixed number of decoding threads of 4.

In Figure 7b, we show the KV-cache content before synchronization at Ⓔ. The naive batched implementation requires duplicating the prefix for each asynchronous thread, which runs as an independent batch item. Since the KV-cache pool is sized for the maximum possible number of parallel threads, many rows often remain unused. This wastes both accelerator memory and computation, as the KV-cache pool's shape determines the shape of attention computation.

In Figure 7c, we show the KV-cache content after synchronization, two decoding steps into Ⓕ. The naive interpreter must copy and insert Fork#1 and Fork#2's asynchronous generations after their corresponding `<promise/>` tags in the main thread's KV-cache row, then mark the rows of terminated threads as uninitialized. This naive batched implementation thus suffers from wasted accelerator memory during memory allocation, wasted computation from the oversized attention operations, and additional overhead from KV-cache movement during synchronization.

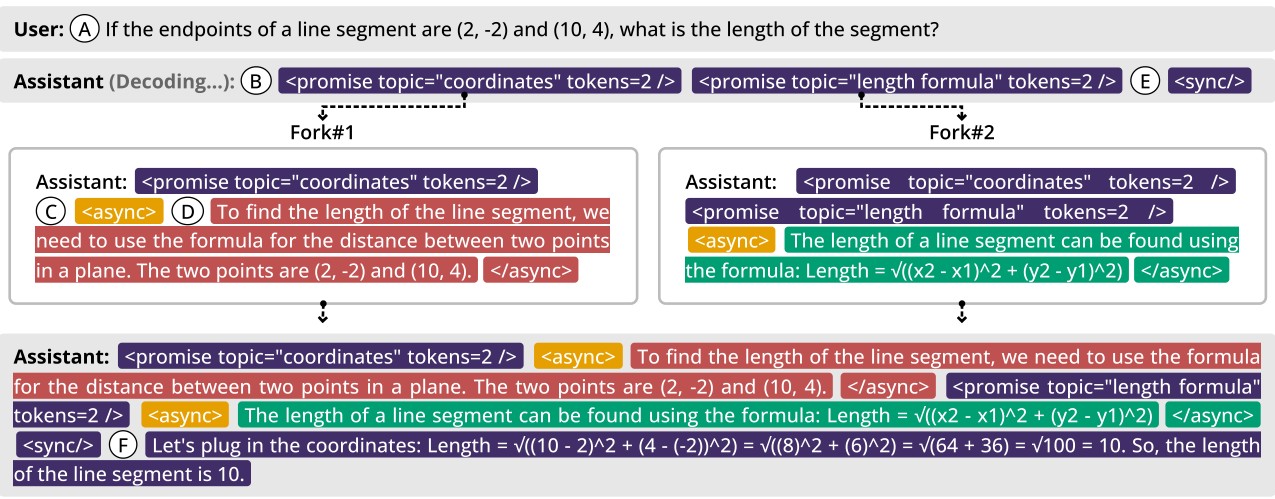

(a) PASTA-LANG interpreter orchestrates parallel decoding. Ⓐ shows the user query. Ⓑ shows the `<promise/>` tag which initiates the first asynchronous decoding thread named "Fork#1". Ⓒ indicates where the interpreter appends an `<async>` tag to the prefix of Fork#1, signaling Fork#1 should complete the promised content with topic "coordinates". Ⓓ denotes the asynchronous generation by Fork#1. Ⓔ shows the `<sync/>` tag where the interpreter pauses to wait for all asynchronous generations. Ⓕ shows the main thread decodes the remaining content with both asynchronous generations in its prefix.

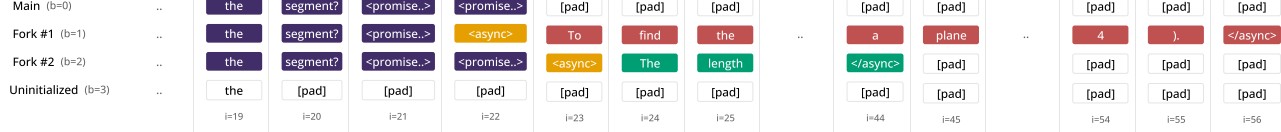

(b) Naive KV-cache layout before synchronization. The naive approach implements asynchronous decoding through batched decoding, with a pre-allocated fixed-size KV-cache pool supporting a maximum number of parallel threads, which is set of 4 in this figure. Each decoding thread operates as an independent batch item, with shared prefix duplicated for each thread. The KV-cache pool is stored in row-major layout with shape (max batch size × sequence length), shown here before synchronization. The figure show KV-cache pool contents immediately after decoding `<sync/>` at Ⓔ.

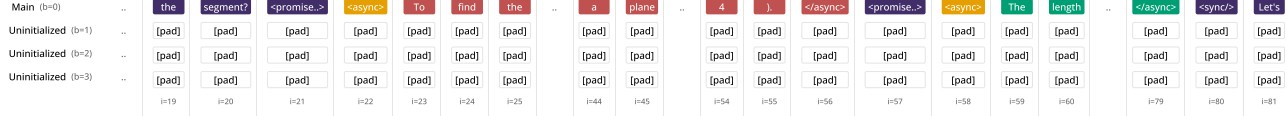

(c) Naive KV-cache layout after synchronization. For asynchronous content to be available to subsequent decoding steps, the naive interpreter must copy and insert each thread's KV-cache (shown in red and green) after their corresponding `<promise/>` tokens in the main thread's KV-cache row. After insertion, the `<sync/>` token is added and subsequent decoding can attend to the asynchronous content. The interpreter must then mark rows corresponding to terminated threads (Fork #1, Fork #2) as uninitialized to free memory. The figure shows the KV-cache pool contents two steps into Ⓕ.

*Figure 7.* Example Naive Interpreter Implementation.

## C. Hyperparamater Selection

We selected the learning rate and epochs that we used to train all models based on which combination of hyperparameters led to the highest quality *Baseline-SFT*-SFT model as measured by length-controlled win-rate on AlpacaEval. We performed a grid search over all combinations of learning rates in $[1e-4, 1e-5, 1e-6]$ and epochs in $[1, 4, 8]$. We found that training with a learning rate of $1e-5$ for $4$ epochs resulting in the highest win-rate model.

## D. PASTA-LANG Preference Score

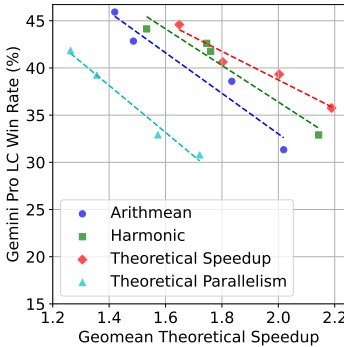

*Figure 8.* Optimization objective. Comparing alternative objectives for preference optimization shows our choice yields the best quality–speedup balance.

When performing PASTA-LANG preference optimization, we compute a preference score for each PASTA-LANG annotated response as a weighted combination of a quality term and a decoding efficiency term, which is meant to reflect the improvement in decoding speed for a response. While our ultimate goal is to increase decoding speed, it is not obvious *a priori* that directly optimizing for the speedup will produce the desired behavior. Such an objective could lead to degenerate solutions such as producing short responses.

**Methodology.** To determine the most performant decoding efficiency term, we investigate four separate metrics for the decoding efficiency of a response: (a) the theoretical speedup of the response only (b) the harmonic mean of the theoretical speedup and theoretical parallelism of the response (c) the arithmetic mean of the theoretical speedup and theoretical parallelism of the response (d) the theoretical parallelism of the response only.

We re-compute the preference score for each training response using each of the efficiency metrics we present above. We then perform a single round of BoNBoN training on each of the different preference labeled datasets.

**Results.** Figure 8 presents the response quality and speedup for each model trained to optimize a different decoding efficiency metric. Intuitively, one might expect the har-

monic mean to be the most effective since it encourages balanced optimization by inducing a larger weight on the weaker metric. However, we find that optimizing for harmonic mean performs similarly to optimizing directly for speedup. We hypothesize this is because our LLM-based quality evaluation naturally favors longer, more detailed responses, preventing the model from artificially increasing speedup through response truncation. As expected, optimizing solely for theoretical parallelism leads to poor speedup, demonstrating the importance of including speedup in the objective. Based on these results, we adopted theoretical speedup only as our efficiency metric.

## E. Evaluation

The prevailing practice in parallel decoding literature uses arithmetic mean to compute average speedup. However, geometric mean should when averaging normalized values such as speedup against a baseline (Fleming & Wallace, 1986). As such, we report geometric mean in Section 5 but include here in Figures 9a to 9c the results computed using arithmetic mean as reference. Notably, by definition, arithmetic mean is larger than or equal to geometric mean, therefore Figures 9a to 9c show notably higher speedup and parallelism than Section 5. Furthermore, PASTA models still Pareto-dominate using arithmetic mean.

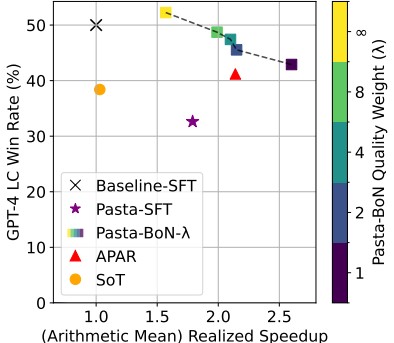

(a) Realized speedup. PASTA models achieve a Pareto-optimal quality–speedup trade-off.

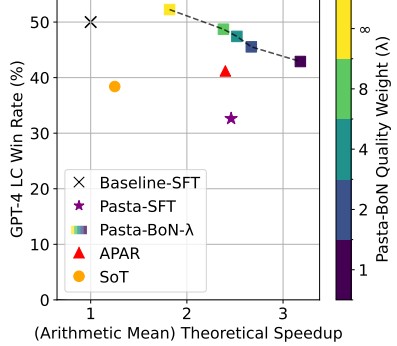

(b) Theoretical speedup. Realized speedup with the PASTA-LANG interpreter is close to the theoretical limit.

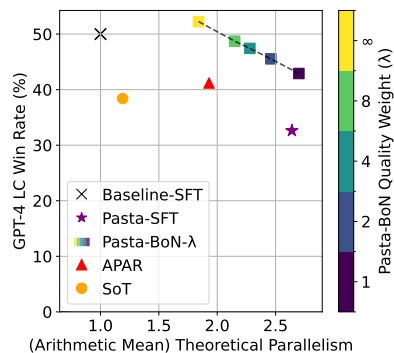

(c) Theoretical parallelism. PASTA outputs exhibit high degree of parallelism.

