# OpenReview forum: "Learning to Keep a Promise: Scaling Language Model Decoding Parallelism with Learned Asynchronous Decoding"
_ICML.cc/2025/Conference — ICML 2025 poster_

### Official Review · Reviewer_9kSW · 2025-03-08

**Overall Recommendation:** 3

**Summary:**

The authors propose a mechanism, Pasta to train large language models (LLMs) to identify and express semantic independence and directly optimizes for both response quality and decoding latency. This framework has two key components, the Pasta-Lang, an annotation language that allows LLMs to express semantic independence, and it also has an interpreter orchestrate parallel decoding on the fly. This two-stage framework achieved state-of-the-art on a benchmark, AlpacaEval, speeding up and improving the quality of generation.

**Claims And Evidence:**

The experiments generally support the claims of the work. However, with only specifying that the work is evaluated on one dataset, the contributions may be limited.

**Essential References Not Discussed:**

I am not aware of any other essential references not discussed.

**Experimental Designs Or Analyses:**

The experiments are adequate. Evaluations on the speed and quality are both provided with enough details.

**Methods And Evaluation Criteria:**

The work only used a benchmark dataset, which may present generalization risks for the proposed method.

**Other Comments Or Suggestions:**

None

**Other Strengths And Weaknesses:**

None

**Questions For Authors:**

None

**Relation To Broader Scientific Literature:**

There’s a generally thorough literature review provided in the paper, and the authors discussed how their research is built on prior work.

**Theoretical Claims:**

The theoretical claims make sense. There’s not much theoretical proofs provided, but more empirical settings.

---

> ### Author Rebuttal · Authors · 2025-04-01
>
> We appreciate the encouraging feedback from the reviewer!
>
> > However, with only specifying that the work is evaluated on one dataset, the contributions may be limited.
>
> We would like to clarify that AlpacaEval actually is a suite of 5 different benchmarking datasets: Self-Instruct [1], Open-Assistant [2], Vicuna [3], Koala [4], and hh-rlhf [5].
>
>
> References:
> [1] Wang, Yizhong, et al. Self-Instruct: Aligning Language Models with Self-Generated Instructions. ACL. 2023. https://arxiv.org/abs/2212.10560
>
> [2] Köpf, Andreas. OpenAssistant Conversations -- Democratizing Large Language Model Alignment. NeurIPS. 2023. https://arxiv.org/abs/2304.07327
>
> [3] Chiang, Wei-Lin, et al. Vicuna: An Open-Source Chatbot Impressing GPT-4 with 90%* ChatGPT Quality. 2023. https://lmsys.org/blog/2023-03-30-vicuna/
>
> [4] Geng, Xinyang, et al. Koala: A Dialogue Model for Academic Research. 2023. https://bair.berkeley.edu/blog/2023/04/03/koala/
>
> [5] Bai, Yuntao, et al. Training a Helpful and Harmless Assistant with Reinforcement Learning from Human Feedback. 2022. https://arxiv.org/abs/2204.05862

---

### Official Review · Reviewer_gQzb · 2025-03-12

**Overall Recommendation:** 4

**Summary:**

This work presents a novel framework that allows the model to learn semantic boundaries in the sequence space, so that semantically independent chunks could be decoded in parallel, or async manner. First, they proposed an XML-based markup to be able to structure a flat sequence into structured chunks that can be used by the decoding controller. Specifically, decoding could switch from usual decoding into async mode, and then synchronize chunks back and use previous chunks as context to decode the final part of the sequence.
Authors design training pipeline so that the model can learn when to generate control tokens to switch into async mode and back. Training pipeline involves both supervised and preference finetuning. Their experimental results show that they method is able to maintain or improve the quality of model responses while making decoding faster due to ability to process async chunks in parallel.

**Claims And Evidence:**

* the overall idea of allowing the model to split the chain of thought into a set of semantically independent chunks is sound and has clear motivation given recent advances in reasoning with long chain of thoughts.
* the benefits of splitting decoding chunks into parallel executions that can synchronize later is an interesting idea, however, the practical utility of such method is questionable especially on the efficiency side. Efficient and optimal async inference of such blocks (while keeping consistent kv cache of the sequential prefix) might be overly complicated usecase unless all major foundational language models would use that.

**Essential References Not Discussed:**

None

**Experimental Designs Or Analyses:**

Experimental design is described clearly. The presented method shows robust improvements in quality and speed compared to other async decoding methods.

**Methods And Evaluation Criteria:**

* experimental testbed uses public benchmarks for performance, which is good, but "theoretical speedup" on decoding side is a bit less convincing.

**Other Comments Or Suggestions:**

* it would help to emphasize that LC winrates in alpaca eval shall be interpreted as "higher-better" , same for all other metrics in the experiments.

**Other Strengths And Weaknesses:**

Overall the biggest concern i have is the potential impact of such approach in the community, I don't see such approach being used widely, but I could be biased. However, I think the proposed method makes sense, and I don't see any flaws in the experimental design!

**Questions For Authors:**

Do you think such approach could bring improvements in model's ability in long chain of thought reasoning? If yes, what would be an essential experiment to do in context of this work to showcase that?

**Relation To Broader Scientific Literature:**

I'm not very familiar with async decoding literature, but related work section looks like it well represents related work, and goes even further into decoding efficiency discussion about methods such as speculative decoding.

**Theoretical Claims:**

* presented speed-ups (geomean theoretical speedup) usually comes with quality degradation, which is a bit concerning given that in theory such async chunks might help to improve reasoning abilities by reducing hallucinations. In that sense the hypotheses suggested by authors do not hold very strong in the experimental part, unless I misunderstood something.

---

> ### Author Rebuttal · Authors · 2025-04-01
>
> We appreciate the reviewer for the helpful comments!
>
> > the benefits of splitting decoding chunks into parallel executions that can synchronize later is an interesting idea, however, the practical utility of such method is questionable especially on the efficiency side. Efficient and optimal async inference of such blocks (while keeping consistent kv cache of the sequential prefix) might be overly complicated usecase unless all major foundational language models would use that.
>
> Our results directly address efficiency. Our implementation achieves practically significant speedups up to 1.9x and on average reaches 78.6% of the theoretical speedup (an optimistic estimate of maximum possible speedup). These results clearly demonstrate real efficiency gains.
> Further, our interpreter design is model-agnostic. Any foundation model can reuse our implementation as long as it adopts the same control tokens to orchestrate asynchronous decoding.
>
> > experimental testbed uses public benchmarks for performance, which is good, but "theoretical speedup" on decoding side is a bit less convincing.
>
> We emphasize that: 1) The left plot in Fig. 3 presents measured wallclock speedup. 2) Theoretical speedup is helpful because end-to-end evaluation entangles the quality of our training algorithm's with the quality of our interpreter implementation. Theoretical speedup lets us examine how effectively our training algorithm promotes speedup, independent of our interpreter implementation.
>
> > presented speed-ups (geomean theoretical speedup) usually comes with quality degradation, which is a bit concerning given that in theory such async chunks might help to improve reasoning abilities by reducing hallucinations. hypotheses suggested by authors do not hold very strong in the experimental part, unless I misunderstood something.
>
> Could the reviewer please clarify why async chunks might help reduce hallucinations and specify what "hypotheses suggested" refers to?
>
> We acknowledge the quality-speedup tradeoff and have been transparent about it throughout our paper, including in the abstract. Our contribution is achieving the best quality-speedup tradeoffs among existing asynchronous decoding methods.
>
> > it would help to emphasize that LC winrates in alpaca eval shall be interpreted as "higher-better" , same for all other metrics in the experiments.
>
> We will add this clarification in the paper.
>
> > Do you think such approach could bring improvements in model's ability in long chain of thought reasoning? If yes, what would be an essential experiment to do in context of this work to showcase that?
>
> We believe mathematical reasoning benchmarks are promising application scenarios for PASTA. However, we also emphasize that DeepSeek R1 (first open source long CoT model) was released only ~8 days prior to ICML deadline. We will include the results of using Pasta with long chain of thought reasoning models in the final paper.

---

> > ### Comment · Reviewer_gQzb · 2025-04-05
> >
> > thanks for addressing my comments!
> >
> > > Our contribution is achieving the best quality-speedup tradeoffs among existing asynchronous decoding methods.
> >
> > this makes sense
> >
> > given the authors desire to add more experiments relevant to long CoT reasoning, Im increasing my score

---

### Official Review · Reviewer_RqHB · 2025-03-13

**Overall Recommendation:** 3

**Summary:**

The paper addresses LLM inference speed and proposes a method to enable asynchronous, parallel decoding of partial LLM responses. The main idea is to fine-tune the model to output special tags for partial answers that are independent of each other and that can then be decoded in parallel. The critical ingredients of the method are:
- Obtaining a dataset which contains markup identifying independent parts of answers; this is achieved by distillation from a powerful LLM (Gemini)
- Training the model to actually produce the independent parts in an independent manner, where the important detail is adjusting position encodings
- Implementing a decoder that can realize the new parallel decoding ability into lower wall-clock time during inference.

**Claims And Evidence:**

The paper claims superior results in both quality and speed compared to existing asynchronous decoding methods; the one considered for comparison is APAR. The authors mention SoT, another relevant baseline, in L234 as a comparison point but I did not find any results with it. After the SFT stage with annotated examples, the model performs worse than APAR in both speedup and quality; after rejection sampling training, the authors achieve higher output quality and higher speed-up compared to APAR, validating the pareto-dominant claim of the abstract.

It should be noted that here, "quality" is measured by win-rates according to LLM-as-a-judge. The authors took some precaution and selected the hyper-parameters of their algorithm with Gemini (Fig. 4, Fig. 5) while performing final evaluations with GPT-4, but comparing models on benchmarks with objective metrics would be appreciated to remove bias.

**Essential References Not Discussed:**

n/a

**Experimental Designs Or Analyses:**

Experimental setup looks good to me; the criteria for the preference tuning stage with BonBon are sensible (finding a trade-off between speed and quality).

I did not like that the analysis promises future improvements from further rounds of BonBon (5.3, "Continuous Improvements") but stops at 2 rounds, while this style of training has trouble achieving benefits after more than 2 iterations (e.g., https://arxiv.org/abs/2308.08998).

**Methods And Evaluation Criteria:**

Yes, quality and speed are indeed the main criteria to evaluate methods that improve decoding speed.

Regard the overall method, I like the idea of a model-driven parallelization strategy, and I can indeed see how for many questions, in particular in combination CoT, answers contain some independent and some dependent parts. What's unfortunate is that the method requires a dedicated training strategy that appears to be pervasive, requiring not only SFT data but also further dedicated fine-tuning stages. It's not clear whether existing fine-tuned models can be easily adapted to follow the format.

The method also relies on specifics regarding the handling of position encodings during parallel decoding. While, in the end, the model does predict the expected length of the response itself, the decision to have it output a multiple of 10 seems arbitrary and to be selected because it turned out to work best. It's unclear to me what the impact of this choice would be if, for example, decoding would be performed with top-p sampling instead of greedy. I am a bit surprised by the noise in Figure 4 and would have expected oracle positions to perform best; here, they perform badly in terms of quality. I would appreciate a general comment regarding noise in the paper's evaluations.

**Other Comments Or Suggestions:**

- Fig. 1 caption, "It wait"
- L135 right, "Pasta-Lang is XML-like"
- L254 left, "BonBon training" (iterations?)

**Other Strengths And Weaknesses:**

While the writing is mostly clear, I found it a bit confusing that the related work is spread throughout the paper. There is an initial discussion in the introduction, then section 2, and finally there is a dedicated related work section which discusses work that, in my opinion, are not very related to this paper. I would prefer a joint discussion in a single section; for example, I don't think the postulated dichotomy is very relevant to the method overall apart from describing related work.

For the "tokens" parameter of the <promise> tag, it took a while to find out what it actually means. It first appears in Fig 1, the main figure presenting the method, without explanation, then in section 3 where is referred to an integer attribute (which makes it appear from Fig. 1 as the model would get them wildly wrong), then finally on page 7 in the discussion of the ablations we learn that it represents 10x the expected token count. Maybe I missed an earlier explanation, but it would be good to clarify this early, e.g., in the figure caption or in the "Annotations." paragraph of the intro.

I would suggest the authors refrain from labeling their method as "pareto-optimal" without adding the qualifier "among the methods that were compared" (e.g., L128 right).

**Questions For Authors:**

- Do you find that the estimated token counts in the promises are accurate?

**Relation To Broader Scientific Literature:**

APAR, the main point of comparison, uses a similar technique to achieve decoding speed-ups but relies on hand-crafted heuristics to identify independent sub-sequences in the LLM output. Instead, this paper asks Gemini to produce an annotated dataset to bootstrap the model's ability to announce parallel sections The method presented here offers more flexibility in structuring the output, and via sampling and preference optimization, the model can be further tuned towards higher speed and quality.

**Theoretical Claims:**

No theoretical claims.

---

> ### Author Rebuttal · Authors · 2025-04-01
>
> We appreciate the reviewers careful reading and insightful comments!
>
> > The authors mention SoT, another relevant baseline, in L234 as a comparison point but I did not find any results with it.
>
> We have included here the updated Figure 3 with the SoT results (please see [rebuttal doc](https://anonymous.4open.science/r/icml-rebuttal-2025-8A52/ICML_Rebuttal.pdf) Figure 1). Notably, we do not observe any speedup by SoT, when applied to Baseline-SFT. We believe that SoT, as a prompt-based method requires the base model to have strong instruction-following ability to perform well, and
> validated this hypothesis by applying SoT to the stronger official instruction-finetuned Gemma-IT model from Google. With this generous implementation, SoT achieves a 1.61x
> speedup while dropping its win rate by 12%. In contrast, our round 2 Pasta-BoN model with quality weight 2 achieves 1.62x speedup with only a 5% drop to win rate.
>
> > comparing models on benchmarks with objective metrics would be appreciated to remove bias.
>
> We thank the reviewer for this suggestion. We will include an objective evaluation benchmark in the final paper.
>
> > Regard the overall method, I like the idea of a model-driven parallelization strategy, and I can indeed see how for many questions, in particular in combination CoT, answers contain some independent and some dependent parts. What's unfortunate is that the method requires a dedicated training strategy that appears to be pervasive, requiring not only SFT data but also further dedicated fine-tuning stages. It's not clear whether existing fine-tuned models can be easily adapted to follow the format.
>
> Our post-training process mirrors RLHF (SFT + preference optimization). PASTA extends RLHF to improve not just quality but also latency. As RL methods take an increasingly central role in LLM training [1], PASTA is well-positioned for integration with these training algorithms.
>
> > the decision to have it output a multiple of 10 seems arbitrary and to be selected because it turned out to work best.
>
> We considered the following candidates of position ID adjustment techniques:
> - Const: assume a fixed async chunk length,
> - Pred-1: have the LLM predict async chunk length in multiples of 1,
> - Pred-10: have the LLM predict async chunk length in multiples of 10.
>
> In addition to our empirical analysis, we have clear rationales for selecting Pred-10. Having the LLM predict async chunk length (rather than using a fixed constant) makes position ID adjustment learnable. We chose multiples of 10 over exact token count prediction (Pred-1) because Pred-1 creates an unrealistic training condition—it predicts length perfectly during training, so the model never encounters position ID collisions. During inference, however, prediction errors inevitably cause such collisions. Pred-10 predicts async chunk length with error due to rounding, intentionally creating imperfect but realistic training conditions that better prepare the model for inference-time position ID collisions.
>
> > It's unclear to me what the impact of this choice would be if, for example, decoding would be performed with top-p sampling instead of greedy.
>
> We swept the following sampling parameters {T=0.5, T=1} x {TopP=0.2, TopP=0.4, TopP=0.8} for each of our 5 round 2 models. Results show similar quality/speedup trade-off as greedy sampling. Please see [rebuttal doc](https://anonymous.4open.science/r/icml-rebuttal-2025-8A52/ICML_Rebuttal.pdf) Figure 2).
>
> > I am a bit surprised by the noise in Figure 4 and would have expected oracle positions to perform best; here, they perform badly in terms of quality. I would appreciate a general comment regarding noise in the paper's evaluations.
>
> The SFT stage shows slightly higher noise (std over 3 runs is 0.6% win rate) while the BoN stage exhibits much less noise. In Figure 3, we report performance across 10 BoN models, providing clear evidence of stability and consistency.
>
> We hypothesize this difference is because we do not optimize for a specific quality/speedup trade-off during SFT, which means the resulting model may land at different points on the same Pareto frontier. Whereas for the BoN stage, we do explicitly optimize for a specific quality/speedup trade-off, making the optimization problem more constrained and the solution more stable.
>
> > Do you find that the estimated token counts in the promises are accurate?
>
> Inspecting the outputs from 5 round 2 models, we find the average relative error to be 14.5%, computed as average(abs(prediction - ground_truth)/ground_truth).
>
> > I did not like that the analysis promises future improvements from further rounds of BonBon (5.3, "Continuous Improvements") but stops at 2 rounds, while this style of training has trouble achieving benefits after more than 2 iterations.
>
> We will include results with 3 rounds of training and update our analysis accordingly in the final paper.
>
> [1] DeepSeek-R1: Incentivizing Reasoning Capability in LLMs via Reinforcement Learning.

---

### Official Review · Reviewer_dvEG · 2025-03-14

**Overall Recommendation:** 4

**Summary:**

This paper proposes PASTA, a method for accelerating LLM inference via promises. It's based on PASTA-lang, a markup-based language that expresses which parts of a prompt might be semantically independent. They write a specification for PASTA-lang, use Gemini-Flash to annotate an instruction-tuning dataset with asynchronous tags in PASTA-lang, and finally finetune a model on PASTA data. The finetuned model achieves significant inference-time speedups on tasks that can be decomposed into semantically independent chunks.

**Claims And Evidence:**

Yes, the benchmarking results are very compelling, as well as the training and inference time measurements.

**Essential References Not Discussed:**

n/a

**Experimental Designs Or Analyses:**

The experimental design was thorough including various ablations, speed analyses, and description of the finetuning process.

**Methods And Evaluation Criteria:**

Yes, the method makes a lot of sense, and the evaluation is thorough and includes Pareto curves for various tradeoffs.

**Other Comments Or Suggestions:**

n/a

**Other Strengths And Weaknesses:**

Strengths:
- Method is simple and works well
- Configuring attention mechanisms in this setup is tricky and important, and has been done well in this case ("Training") in Section 4
- Although they only use the SlimOrca dataset, the finetuning setup seems general enough that it might be reusable in other domains
- Experimental analysis is thorough and interesting

Weaknesses:
- How generalizable is this method? Does finetuning on SlimOrca in PastaLang produce generalization capabilities that would help for other unrelated tasks/domains?
- How does PASTA compare to speculative decoding? Could they be used in parallel, with speculative decoding active within each asynchronous chunk?

**Questions For Authors:**

- Why doesn't Figure 3 compare directly to skeleton-of-thought?
- Even though it's synchronous, shouldn't standard speculative decoding be a suitable baseline as well?

**Relation To Broader Scientific Literature:**

This paper is a nice contribution to the LLM inference literature, specifically under the umbrella of asynchronous decoding. It does require special tuning on a specific dataset, so it is not adaptable to the general case, but it's a nice proof-of-concept.

**Theoretical Claims:**

n/a

---

> ### Author Rebuttal · Authors · 2025-04-01
>
> We thank the reviewer for the encouraging feedback and thoughtful comments! Here're our responses.
>
> > How generalizable is this method? Does finetuning on SlimOrca in PastaLang produce generalization capabilities that would help for other unrelated tasks/domains?
>
> We designed our method and evaluation to accommodate a wide variety of tasks/domains. Specifically, SlimOrca was derived from the FLAN collection of instruction-following dataset, covering 1800+ tasks[2]. Furthermore, our evaluation also covers a wide range of representative real world interaction with LLM including translation/summarization/explanation/creative writing/mathematical problem solving/coding. As [1] notes:
> “[AlpacaEval] operates on a fixed set of 805 instructions chosen to be representative of user interactions on the Alpaca web demo.”
> Therefore, our results demonstrate generalizations across a considerable range of tasks/domains already.
>
> > How does PASTA compare to speculative decoding? Could they be used in parallel, with speculative decoding active within each asynchronous chunk?
>
> > Even though it's synchronous, shouldn't standard speculative decoding be a suitable baseline as well?
>
> We indeed believe PASTA and speculative decoding are complementary techniques that compose well. As you mentioned, speculative decoding can accelerate each asynchronous chunk in PASTA, creating a multiplicative speedup effect. We will include the results of using speculative decoding in the final paper.
>
> > Why doesn't Figure 3 compare directly to skeleton-of-thought?
>
> We include here the updated Figure 3 with skeleton-of-thought results (please see [rebuttal doc](https://anonymous.4open.science/r/icml-rebuttal-2025-8A52/ICML_Rebuttal.pdf) Figure 1). As described in Section 5, we do not observe speedup by SoT, when applied to Baseline-SFT. We believe that SoT, as a prompt-based method, requires the base model to have strong instruction-following ability to perform well, and validated this hypothesis by applying SoT to the stronger official instruction-finetuned Gemma-IT model from Google. With this generous implementation, SoT achieves a 1.61x speedup while dropping its win rate by 12%. In contrast, our round 2 Pasta-BoN model with quality weight 2 achieves 1.62x speedup with only a 5% drop to win rate.
>
> References:
>
> [1] Dubois, Y., et al. Length-controlled alpacaeval: A simple way to debias automatic evaluators. arXiv preprint arXiv:2404.04475, 2024
>
> [2] Longpre, S., et al. The Flan Collection: Designing Data and Methods for Effective Instruction Tuning. arXiv preprint arXiv:2301.13688, 2023.

---

### Decision · Program_Chairs · 2025-05-01

**Decision:**

Accept (poster)

**Comment:**

This paper proposes PASTA, a method for accelerating LLM inference via parallel autoregressive generation, when it is possible to do so. The proposed method appears to be simple and works well. All reviewers praised the significance of the method. It is however, a bit complex as it requires specialized fine-tuning and engineering, which can limit it being widely adopted. Nonetheless, I recommend acceptance